# Exogenous Glutathione Promotes the Proliferation of *Pinus koraiensis* Embryonic Cells and the Synthesis of Glutathione and Ascorbic Acid

**DOI:** 10.3390/plants11192586

**Published:** 2022-09-30

**Authors:** Fang Gao, Yujie Shi, Ruirui Wang, Iraida Nikolaevna Tretyakova, Alexander Mikhaylovich Nosov, Hailong Shen, Ling Yang

**Affiliations:** 1State Key Laboratory of Tree Genetics and Breeding, School of Forestry, Northeast Forestry University, Harbin 150040, China; 2State Forestry and Grassland Administration Engineering Technology Research Center of Korean Pine, Harbin 150040, China; 3Laboratory of Forest Genetics and Breeding, V.N. Sukachev Institute of Forest, Siberian Branch of RAS, Krasnoyarsk 660036, Russia; 4Department of Cell Biology, Institute of Plant Physiology K.A. Timiryazev Russian Academy of Sciences, Moscow 127276, Russia; 5Department of Plant Physiology, Biological Faculty, Lomonosov Moscow State University, Moscow 119991, Russia

**Keywords:** Korean pine, embryonic callus, proliferative capacity, glutathione, ASCORBIC acid

## Abstract

Somatic embryogenesis (SE), which leads to the formation of embryonic callus (EC) tissue, is the most promising method for large-scale production and selective breeding of woody plants. However, in many species, SE suffers from low proliferation rates, hindering the production of improved plant materials. One way of improving proliferation rates is achieved by improving the redox status of the culture medium. In this study, we investigated the effects of exogenous glutathione (GSH) and L-buthionine sulfoximine (BSO, the inhibitor of glutathione synthase) on the EC proliferation rate in Korean pine (*Pinus koraiensis*), using cell lines with both high (F: 001#-001) and low (S: 001#-010) proliferation potential. We found that exogenous GSH promoted cell proliferation in both cell lines, while exogenous BSO inhibited proliferation in both cell lines. At 35 d with exogenous GSH treatment, the fresh weight of F and S cell lines increased by 35.48% and 48.39%, respectively, compared with the control. The exogenous application of GSH increased the intracellular levels of GSH, total GSH (T-GSH), oxidized glutathione (GSSG), ascorbic acid (ASA), total ASA (T-ASA), and the ratios of GSH:T-GSH and ASA:T-ASA in both F and S cell lines. Furthermore, exogenous GSH increased the activity of both glutathione reductase (GR) and dehydroascorbate reductase (DHAR) while decreasing the activity of ascorbate peroxidase (APX) in both cell lines. It appears that the application of exogenous GSH promotes a reducing cultural environment, which is conducive to EC proliferation in Korean pine. By helping to reveal the mechanism whereby GSH regulates redox homeostasis in Korean pine EC cells, we have laid the foundation for a large-scale breeding of Korean pine somatic embryogenesis technology system.

## 1. Introduction

Somatic embryogenesis (SE) is the artificial formation of embryo-like structures from somatic cells without the need for gametic fusion [1]. SE is often used as a model system for studying the physiological, biochemical, and molecular processes of embryonic development [2,3,4,5]. Additionally, because SE offers the advantages of genetic stability and rapid reproduction [6], this technique is particularly well-suited to the selective breeding of woody plants [7]. In large-scale tree propagation, SE allows for the fast and efficient production of a large number of propagules through bypassing the reproductive process. Using SE can possibly accelerate the breeding process by around 20–30 years in comparison with seed orchards [8,9]. Conifer species in particular must be propagated through indirect SE, which involves the production of undifferentiated cells which must undergo embryonic callus (EC) formation. Unfortunately, indirect SE of conifers can be problematic for large-scale production due to the low proliferation rate of EC [10].

Korean pine (*Pinus koraiensis* Sieb. et Zucc.) is a dominant species in mixed-broadleaved Korean pine forests in Northeast China [11,12]. It is a conifer with particular economic importance in producing high-quality wood for construction, furniture, and flooring. The bark, pine needles, and cone scales are also important raw materials for the production of volatile oils and turpentine [13,14]. Although the Korean pine SE technique has been established [12,15,16,17], many problems and shortcomings remain; for example, the proliferation potential of different cell lines varies greatly, making SE in this species inefficient for large-scale breeding efforts.

Cellular antioxidants such as glutathione (GSH)/oxidized glutathione (GSSG) and ascorbic acid (ASA)/dehydroascorbic acid (DHA) play an important role in many biological processes, including cell division and differentiation, and can impact the process of SE through changes in cellular reduction oxidation (redox) status [18,19,20,21]. For example, GSH increased the relative growth in the EC of date palm (*Phoenix dactylifera*) [22]. In white spruce (*Picea glauca*), regulation of the redox status of the SE medium by application of exogenous GSH or GSSG promoted a dramatic increase in both EC proliferation (25% by weight) and the number of early filamentous embryos, accompanied by an increase in ATP and nucleotide production. Inhibiting GSH synthesis through the application of L-buthionine sulfoximine (BSO) has been shown to induce a pro-oxidative environment, thereby reducing the cellular GSH: total glutathione (T-GSH) ratio [23]. Both ASA and GSH are core components of the ASA-GSH cycle, and in plants, they are produced primarily by glutathione reductase (GR), dehydroascorbate reductase (DHAR), and ascorbate peroxidase (APX). However, the effects of GSH and BSO on Korean pine EC proliferation are unknown.

The proliferative capacity of EC in conifers is essential for the large-scale production of high-quality seedlings. GSH promotes EC proliferation in white spruce, but it is unclear whether GSH has an effect on EC proliferation in Korean pine. The effect of exogenous GSH on the accumulation of GSH and ASA in EC cells of Korean pine and the cycling between them is not clear. In this study, we investigated the effects of exogenous GSH and BSO on the EC proliferation rate in Korean pine, using cell lines with both high (F: 001#-001) and low (S: 001#-010) proliferation potential. Furthermore, we examined redox regulation, as well as intracellular GSH and ASA synthesis, in these cell lines. This work will serve to optimize large-scale breeding efforts in woody plants by improving SE efficiency, particularly in Korean pine and other conifer species.

## 2. Results

### 2.1. Effects of Exogenous GSH and BSO on EC Proliferation

The growth status of the two cell lines is shown in Figure 1 (Figure 1a for F cell line, and Figure 1b for S cell line).The effects of exogenous application of GSH and BSO on EC fresh weight (FW) of the two cells lines were different (*p* < 0.05) (Figure 2). Exogenous GSH promoted cell proliferation in both cell lines (Figure 2a), whereas exogenous BSO inhibited cell proliferation in both cell lines (Figure 2b). The FW of the EC in the F cell line increased rapidly during 7–21 d with exogenous GSH treatment, while the growth gradually slowed down during 21–35 d (Figure 2a). At 35 d of proliferation culture, the FW of the F cell line treated with GSH was 2.52 g, which was 35.48% higher than that of the control (1.86 g). The FW growth trend of the S cell line after exogenous addition of GSH and BSO was the same as that of the F cell line, but the FW of the S cell line was lower than that of the F cell line throughout the culture period. At 35 d of proliferation culture, the FW of the S cell line treated with GSH was 1.84 g, which was 48.39% higher than that of the control (1.24 g). Thus, exogenous GSH treatment resulted in greater FW growth in the S cell line than in the F cell line.

The effects of exogenous application of GSH and BSO on EC dry weight (DW) of the two cells lines were different (*p* < 0.05) (Figure 3). Overall, exogenous GSH promoted the increase in DW in both cell lines (Figure 3a), while exogenous BSO inhibited this increase (Figure 3b). The DW of F cell lines increased rapidly from 7 to 21 d with GSH treatment, while DW growth slowed down from 21 to 35 d (Figure 3a). The DW of F cell lines treated with GSH was 0.153 g at 35 d of proliferation culture, which was 31.89% higher than that of the control (0.116 g). The growth trend of DW of the S cell line after exogenous addition of GSH and BSO was the same as that of the F cell line, but the DW of the S cell line was lower than that of the F cell line overall. The dry weight of S-GSH was 0.104 g at 35 d of proliferation culture, which was 48.57% higher than that of the control (S-CK EC, 0.070 g). Thus, exogenous GSH treatment resulted in greater DW growth in the S cell line than in the F cell line.

**Figure 3 plants-11-02586-f003:**
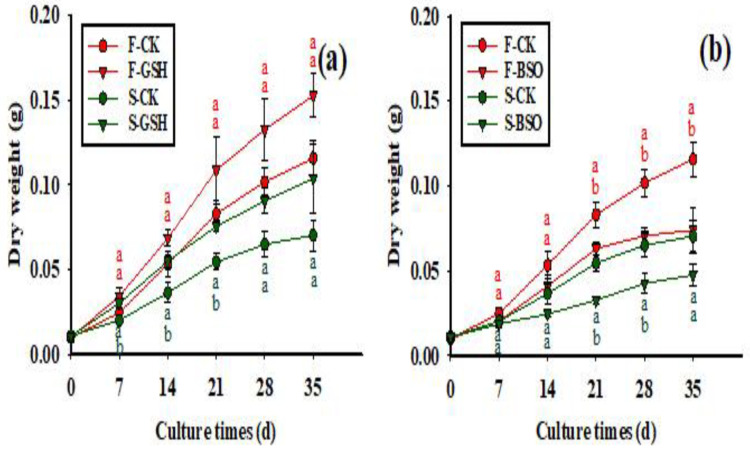
Effects of exogenous application of GSH and BSO on Korean pine EC DW. Note: (**a**) shows the CK and GSH treatment, (**b**) shows the CK and BSO treatment. Different lowercase letters at the same culture time indicate significant differences (*p* ˂ 0.05). Red lowercase letters indicate F cell line GSH, CK, BSO treatment, and green lowercase letters indicate S cell line GSH, CK, BSO treatment. The average EC weight of five Petri dishes was counted for each treatment. ANOVA and Duncan’s test were performed on the data (mean ± se) in the figure.

The relative water content of the F cell line across treatments tended to increase at first and then decrease as culture time progressed (Figure 4a,b), reaching a peak at day 7. The S cell line showed a similar trend to the F cell line, reaching a relative water content of 94.19% for the S-CK treatment, 94.38% for the S-GSH treatment (Figure 4a), and 94.53% for the S-BSO treatment (Figure 4b) by 35 days. Overall, there was no significant difference in relative water content between the two cell lines across the three treatments (*p* > 0.05) (Figure 4).

**Figure 4 plants-11-02586-f004:**
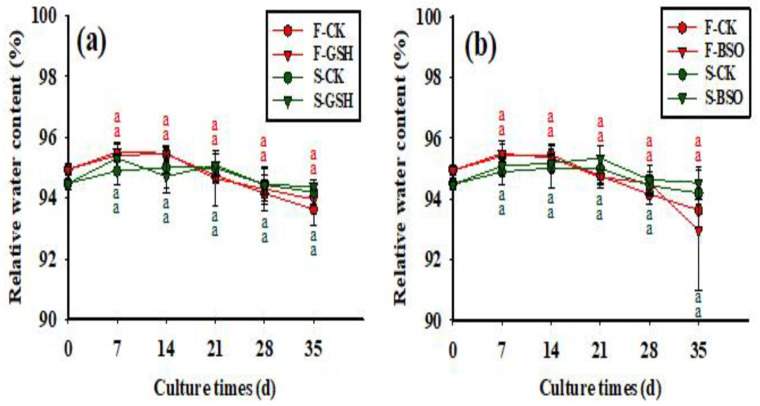
Effects of exogenous application of GSH and BSO on Korean pine EC relative water content. Note: (**a**) shows the CK and GSH treatment, (**b**) shows the CK and BSO treatment. Different lowercase letters at the same culture time indicate significant differences (*p* ˂ 0.05). Red lowercase letters indicate F cell line GSH, CK, BSO treatment, and green lowercase letters indicate S cell line GSH, CK, BSO treatment. The average EC relative water content of five Petri dishes was counted for each treatment. ANOVA and Duncan’s test were performed on the data (mean ± se) in the figure.

### 2.2. Effects of Exogenous GSH and BSO on Intracellular GSH Synthesis

The exogenous application of GSH promoted T-GSH synthesis in both cell lines (Figure 5a,b). In the F-GSH treatment, intracellular T-GSH peaked at day 7 (1.10 μmol/g FW), and then decreased gradually to 0.53 μmol/g FW by day 35 (Figure 5a). The exogenous application of BSO inhibited T-GSH synthesis in both cell lines (Figure 5b). In the F-BSO treatment, intracellular T-GSH decreased from 0 to 35 days, reaching a minimum of 0.02 μmol/g FW by the end of the experiment. The T-GSH content of S and F cell lines showed similar trends after application of exogenous GSH and BSO, although the T-GSH content of F cell lines was always higher than that of S cell lines.

**Figure 5 plants-11-02586-f005:**
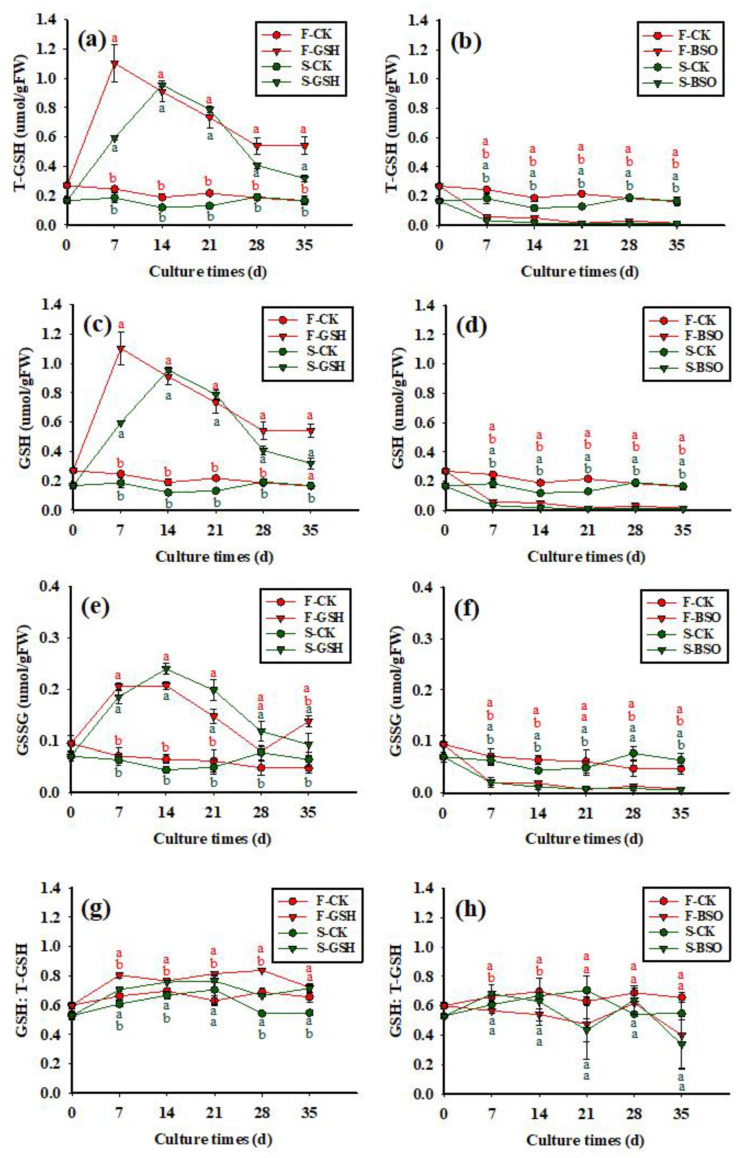
Effects of exogenous application of GSH and BSO on Korean pine EC intracellular GSH synthesis. Note: (**a**,**c**,**e**,**g**) shows the CK and GSH treatment, (**b**,**d**,**f**,**h**) shows the CK and BSO treatment. Different lowercase letters at the same culture time indicate significant differences (*p* ˂ 0.05). Red lowercase letters indicate F cell line GSH, CK, BSO treatment, and green lowercase letters indicate S cell line GSH, CK, BSO treatment. The average glutathione content of three was counted for each treatment. ANOVA and Duncan’s test were performed on the data (mean ± se) in the figure.

The effects of exogenous application of GSH and BSO on the intracellular GSH content of the two cell lines were different (*p* < 0.05) (Figure 5c,d). The exogenous application of GSH promoted intracellular GSH synthesis in the F cell line, with intracellular GSH content rapidly increasing to 0.892 μmol/g FW between 0 and 7 days, and slowly decreasing to 0.393 μmol/g FW by 35 days (a 56% reduction) (Figure 5c). The exogenous application of BSO inhibited intracellular GSH synthesis in the F cell line, with intracellular GSH content gradually decreasing to a minimum value of 0.010 μmol/g FW by day 35 (Figure 5d). The intracellular GSH content of S and F cell lines showed similar trends after application of exogenous GSH and BSO.

The effects of exogenous application of GSH and BSO on the intracellular GSSG content of the two cell lines were different (*p* < 0.05) (Figure 5e,f). Exogenous application of GSH promoted the synthesis of intracellular GSSG in F cell lines, with intracellular GSSG content increasing to 0.207 μmol/g FW between 0 and 14 days, decreasing to 0.080 μmol/g FW from 14 to 28 days, and then slowly increasing again to 0.137 μmol/g FW by 35 days (Figure 5e). The exogenous application of BSO inhibited intracellular GSSG synthesis, with the intracellular GSSG content gradually decreasing to a minimum value of 0.007 μmol/g FW by day 35 (Figure 5f). The intracellular GSSG content of S and F cell lines showed similar trends after application of exogenous GSH and BSO, although the GSSG content of F cell lines was always higher than that of S cell lines.

The exogenous application of GSH increased the intracellular GSH:T-GSH ratio in F and S cell lines (Figure 5g,h), with the intracellular GSH:T-GSH ratio increasing rapidly to 0.81 between 0 and 7 days, and fluctuating until the end of the experiment, with a ratio of 0.74 by day 35. The exogenous application of BSO decreased the intracellular GSH:T-GSH ratio in F cell lines, with the intracellular GSH:T-GSH ratio decreasing gradually to a minimum value of 0.59 by day 35 (Figure 5h). The intracellular GSH:T-GSH ratio was higher in the F-CK treatment than in the S-CK treatment over the course of the experiment (Figure 5g).

### 2.3. Effects of Exogenous GSH and BSO on Intracellular ASA Synthesis

The exogenous application of GSH promoted the synthesis of intracellular T-ASA in F and S cell lines (Figure 6a), with intracellular T-ASA content increasing rapidly to 0.66 μmol/g FW between 0 and 7 days, and then fluctuating until the end of the experiment, with a content of 0.28 μmol/g FW by day 35. The exogenous application of BSO inhibited the synthesis of intracellular T-ASA in F cell lines, with the intracellular T-ASA content decreasing gradually to a minimum value of 0.26 μmol/g FW by day 35 (Figure 6b). The exogenous application of GSH promoted the synthesis of intracellular T-ASA in S cell lines, with the intracellular T-ASA content increasing rapidly to 0.44 μmol/g FW between 0 and 7 days, and then decreasing gradually to a minimum value of 0.17 μmol/g FW by day 35. Similar to the F cell lines, the exogenous application of BSO inhibited the synthesis of intracellular T-ASA in S cell lines, with the intracellular T-ASA content decreasing gradually to a minimum value of 0.16 μmol/g FW by day 35. The intracellular T-ASA content of F cell lines was always higher than that of S cell lines.

**Figure 6 plants-11-02586-f006:**
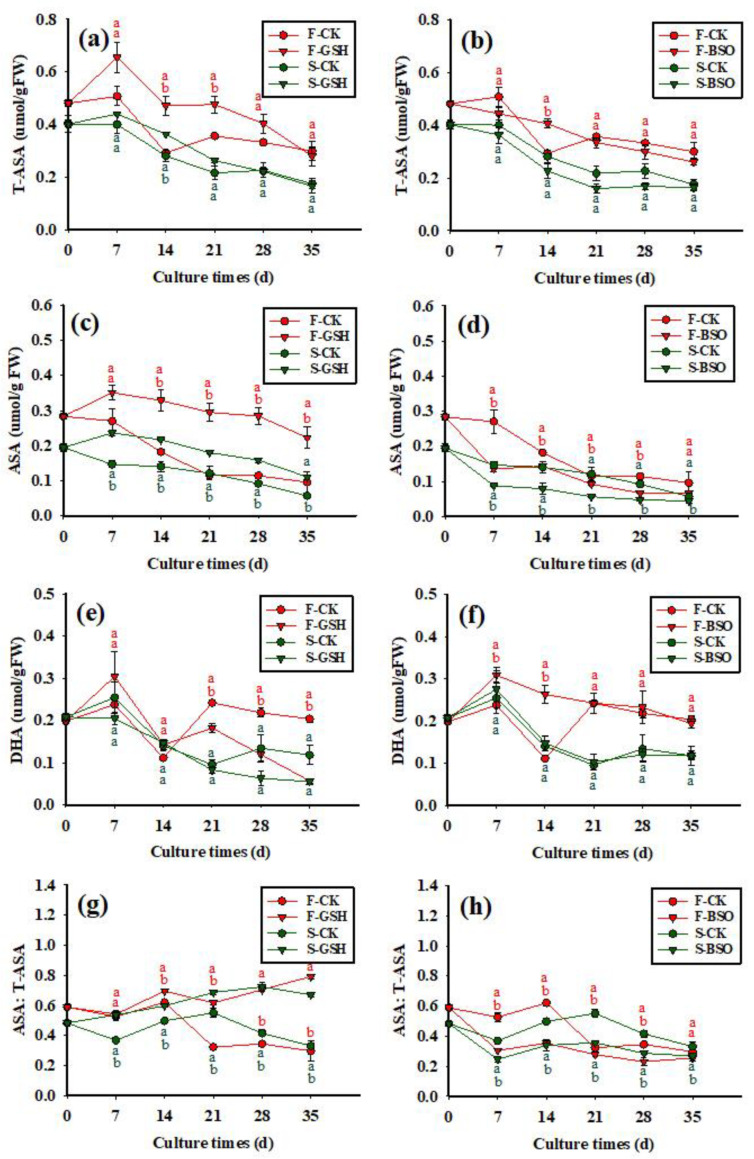
Effects of exogenous application of GSH and BSO on Korean pine EC intracellular ASA synthesis. Note: (**a**,**c**,**e**,**g**) shows the CK and GSH treatment, (**b**,**d**,**f**,**h**) shows the CK and BSO treatment. Different lowercase letters at the same culture time indicate significant differences (*p* ˂ 0.05). Red lowercase letters indicate F cell line GSH, CK, BSO treatment, and green lowercase letters indicate S cell line GSH, CK, BSO treatment. The average ascorbic acid content of four was counted for each treatment. ANOVA and Duncan’s test were performed on the data (mean ± se) in the figure.

The effects of exogenous application of GSH and BSO on the intracellular ASA content of the two cell lines were different (*p* < 0.05) (Figure 6c,d). The intracellular ASA content of the F-CK treatment decreased gradually between 0 and 35 days, reaching a minimum value of 0.10 μmol/g FW by day 35 (Figure 6c). The intracellular ASA content of the F-GSH treatment peaked at day 7 (0.35 μmol/g FW) and decreased slowly to a minimum value of 0.22 μmol/g FW by day 35. The exogenous application of BSO inhibited the synthesis of intracellular ASA in F cell lines (Figure 6d), with the ASA content decreasing rapidly between 0 and 7 days (0.14 μmol/g FW) and fluctuating until the end of the experiment, with a content of 0.07 μmol/g FW by day 35. The intracellular ASA content of F cell lines was always higher than that of S cell lines.

The effects of exogenous application of GSH and BSO on the intracellular DHA content of the two cell lines were differed (*p* < 0.05) (Figure 6e,f). The exogenous application of GSH promoted the synthesis of intracellular DHA in F cell lines (Figure 6e), with the intracellular DHA content increasing rapidly to 0.30 μmol/g FW between 0 and 7 days, and then fluctuating until the end of the experiment, with a content of 0.06 μmol/g FW by day 35. The exogenous application of BSO also promoted the synthesis of intracellular DHA in F cell lines (Figure 6f), with the DHA content increasing rapidly between 0 and 7 days (0.31 μmol/g FW), and then decreasing gradually until the end of the experiment, reaching a minimum value of 0.19 μmol/g FW by day 35. The exogenous application of GSH inhibited the synthesis of intracellular DHA in S cell lines, with the intracellular DHA content gradually decreasing between 0 and 35 days, reaching a minimum value of 0.06 μmol/g FW by day 35. The exogenous application of BSO promoted the synthesis of DHA in S cell lines, with the intracellular content of DHA increasing rapidly from 0 to 7 days (0.28 μmol/g FW), and then fluctuating until the end of the experiment, reaching a value of 0.12 μmol/g FW by day 35. Overall, in F cell lines, the exogenous application of BSO promoted intracellular DHA synthesis, while the exogenous application of GSH promoted intracellular DHA synthesis from 0 to 14 days and inhibited it from 14 to 35 days. In S cell lines, the exogenous application of BSO promoted intracellular DHA synthesis, while the exogenous application of GSH inhibited intracellular DHA synthesis.

The exogenous application of GSH tended to increase the intracellular ASA:T-ASA ratio in F cell lines (Figure 6g), although the ratio fluctuated throughout the experiment, reaching a value of 0.80 by day 35. The exogenous application of BSO decreased the intracellular ASA:T-ASA ratio in F cell lines (Figure 6h), again fluctuating throughout the experiment and reaching a value of 0.26 by day 35. Both the S and F cell lines showed similar patterns of change over time, with exogenous application of GSH tending to increase the intracellular ASA:T-ASA ratio and that of BSO tending to decrease the intracellular ASA:T-ASA ratio in both S and F cell lines.

### 2.4. Effects of Exogenous GSH and BSO on Enzymatic Activity

The effects of exogenous application of GSH and BSO on intracellular GR activity of the two cell lines were differed (*p* < 0.05) (Figure 7a,b). Intracellular GR activity was always higher in F cell lines than S cell lines (Figure 7a). Intracellular GR activity in both the F-CK and S-CK treatments increased from 0 to 7 days (0.121 and 0.060 U/g FW, respectively), decreasing to 0 U/g FW by day 35. Intracellular GR activity in the F-GSH treatment increased from 0 to 7 days (0.193 U/g FW), and gradually decreased to 0.004 U/g FW by day 35 (Figure 7a). The exogenous application of BSO inhibited the intracellular GR activity in the F cell line, with intracellular GR activity gradually decreasing to 0 U/g FW by day 35 (Figure 7b). Intracellular GR activity in the S-GSH treatment peaked at day 7 (0.100 U/g FW), gradually decreasing to 0 U/g FW by day 35.

**Figure 7 plants-11-02586-f007:**
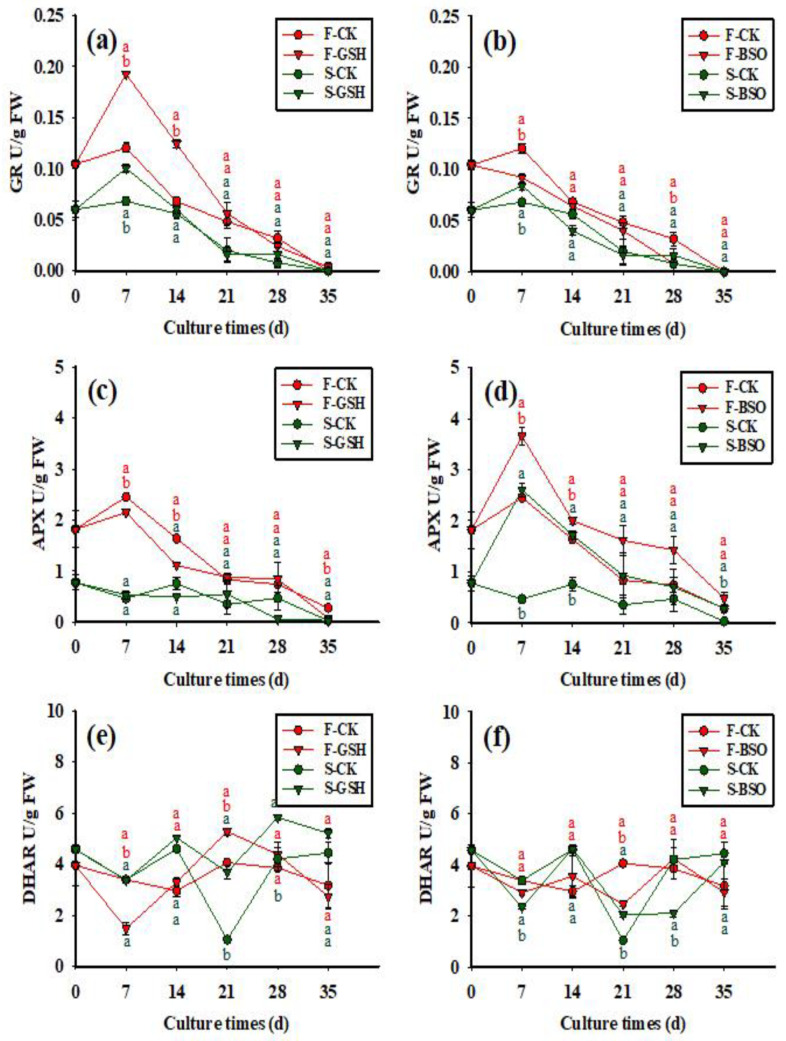
Effects of exogenous application of GSH and BSO on Korean pine EC intracellular GR, APX, and DHAR activity. Note: (**a**,**c**,**e**) shows the CK and GSH treatment, (**b**,**d**,**f**) shows the CK and BSO treatment. Different lowercase letters at the same culture time indicate significant differences (*p* ˂ 0.05). Red lowercase letters indicate F cell line GSH, CK, BSO treatment, and green lowercase letters indicate S cell line GSH, CK, BSO treatment. The average GR content of four was counted for each treatment, APX and DHAR were three. ANOVA and Duncan’s test were performed on the data (mean ± se) in the figure.

The effects of exogenous application of GSH and BSO on intracellular APX activity of the two cell lines were differed (*p* < 0.05) (Figure 7c,d). Intracellular APX activity was always higher in F cell lines than S cell lines (Figure 7c). At day 0, intracellular APX activity was 1.82 μmol/g FW in the F-CK treatment and 0.78 μmol/g FW in the S-CK treatment. Intracellular APX activity in the F-GSH treatment increased from 0 to 7 days (2.15 U/g FW), and gradually decreased to 0.08 U/g FW by day 35. The exogenous application of BSO increased the intracellular APX activity in the F cell line (Figure 7d), with intracellular APX increasing rapidly between 0 and 7 days (3.66 U/g FW), and then decreasing gradually until the end of the experiment, reaching a value of 0.48 U/g FW by day 35. Intracellular APX activity in the S-GSH treatment decreased from 0 to 14 days, then fluctuated until the end of the experiment, reaching a value of 0.03 U/g FW by day 35. Exogenous application of BSO increased the intracellular APX activity in S cell lines.

The effects of exogenous application of GSH and BSO on intracellular DHAR activity of the two cell lines were differed (*p* < 0.05) (Figure 7e,f). In the F-GSH treatment, intracellular DHAR activity decreased from 0 to 7 days (1.48 U/g FW), and then fluctuated until the end of the experiment, reaching a value of 2.74 U/g FW by day 35 (Figure 7e). Similarly, in the F-BSO treatment, intracellular DHAR activity decreased from 0 to 7 days (2.92 U/g FW), and then fluctuated until the end of the experiment, reaching a value of 2.93 U/g FW by day 35 (Figure 7f). Intracellular DHAR activity was higher in S cell lines than F cell lines. The intracellular DHAR activity in S-GSH cells was higher than that in S-CK cells throughout the experiment. In the S-GSH treatment, intracellular DHAR activity decreased from 0 to 7 days (3.37 U/g FW), and then fluctuated until the end of the experiment, reaching a value of 5.22 U/g FW by day 35. In the S-BSO treatment, intracellular DHAR activity decreased from 0 to 7 days (2.37 U/g FW), and then fluctuated until the end of the experiment, reaching a value of 4.11 U/g FW by day 35.

### 2.5. qRT-PCR Validation of Differentially Expressed Genes between GSH and BSO Treatments

Ten differentially expressed Korean pine genes were selected for qRT-PCR validation (Figure 8). Of these, the relative expression of eight genes was higher under GSH treatment compared to BSO treatment. Genes upregulated by exogenous GSH treatment included the GST family (regulating GSH S-transferase) genes c180344.graph_c0, c181305.graph_c0, c187045. graph_c0, c187964.graph_c1, and c188830.graph_c0; the DHAR family (regulating GSH dehydrogenase) gene c183265.graph_c0; the ODC1 family (regulating ornithine decarboxylase) gene 190620.graph_c2; and the GULO family (regulating L-gulonolactone oxidase) genes c187768.graph_c0 and c190620.graph_c2.

**Figure 8 plants-11-02586-f008:**
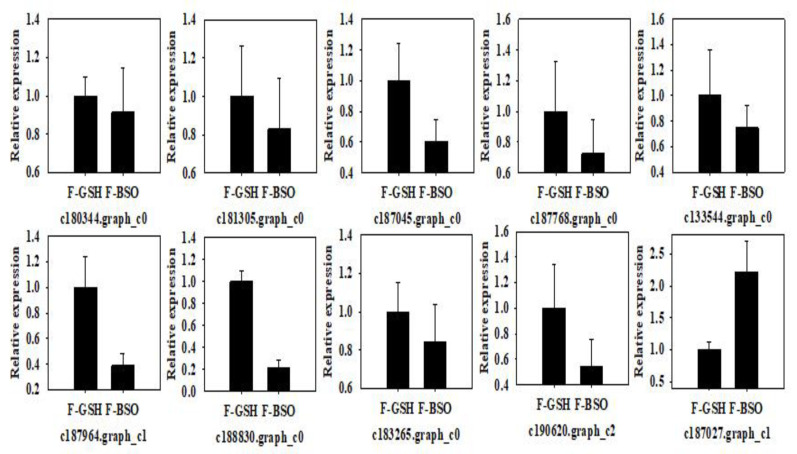
qRT-PCR analysis of glutathione-related genes in Korean pine F and S cell lines.

## 3. Discussion

EC material allows for efficient genetic transformation, makes an ideal system for studying single-cell differentiation and allotropic expression, and can form the basis for large-scale production of woody plants [24]^.^ Healthy EC tissue and a fast proliferation rate are essential during SE. To that end, GSH has been shown to regulate cell division and differentiation in many organisms, and has been found in high concentrations in rapidly proliferating cells [25]. More specifically, GSSG promotes initial cell proliferation, while GSH induces sequential division of EC cells, producing more filamentous embryos through cleavage of polyembryos [26]. Studies of white spruce (*P. glauca*) SE have shown that exogenous application of GSH promotes the proliferation of EC, increasing EC FW by as much as 25%. In this work, we examined the effect of exogenous application of GSH and BSO on Korean pine SE and found that GSH promoted EC proliferation, while BSO inhibited EC proliferation, compared to CK.

GSH is present in most prokaryotic and eukaryotic cells [27] and plays an important role in cellular redox homeostasis as well as cell division and differentiation [10,19]. During the ASA-GSH cycle, DHAR converts GSH to GSSG, which can be converted back to GSH by GR. An increase in the GSH:T-GSH ratio is necessary to effectively prevent reactive oxygen species (ROS) accumulation [28]. In this work, we found that exogenous application of GSH during Korean pine EC proliferation increased intracellular GR activity, GSH and GSSG synthesis, T-GSH content, and the GSH:T-GSH ratio. Both the intracellular GSH:T-GSH ratio and GSH content were higher in the F-CK than in the S-CK cell line, and the proliferation potential of the F cell line was higher than that of the S cell line. Taken together, these results suggest that the proliferation potential of Korean pine EC is related to the intracellular GSH content.

ASA is the primary component of the ASA-GSH cycle and plays an important role in cellular growth, elongation, and division frequency, and also in protecting the cell membrane [29]. In tobacco (*Nicotiana tabacum*), exogenous application of ASA during cell division has been shown to increase intracellular soluble protein content, and decrease both malondialdehyde (MDA) content and cell membrane permeability [30]. ASA is oxidized by APX to form DHA, which can be converted back to ASA via the ASA-GSH cycle. In this work, we found that the exogenous application of GSH promoted the synthesis of T-ASA and ASA, while the exogenous application of BSO promoted the synthesis of DHA. These results suggest that the exogenous addition of GSH both promotes ASA synthesis and inhibits DHA synthesis, thus increasing the ASA:T-ASA ratio.

GSH is an electron receptor for DHA and plays an important role in scavenging intracellular ROS and maintaining redox homeostasis in plants [31]. In both angiosperm and gymnosperm embryogenesis, a reducing cultural environment is required during the early stages of embryogenesis, while an oxidizing environment is required during late embryonic development [32,33,34,35]. It has been suggested that the ratio of GSH to T-GSH is more important than the actual amounts of GSH and GSSG present [36]. In this work, we found that the intracellular GSH content was higher than the GSSG content during EC proliferation, suggesting that GSH plays a dominant role in EC proliferation in Korean pine. Additionally, the exogenous application of GSH to the culture medium resulted in an increase in both the intracellular GSH:T-GSH ratio and the ASA:T-ASA ratio, suggesting that early EC proliferation in Korean pine does benefit from a reducing environment. It also seems clear that the intracellular ASA-GSH cycle is weakened in later EC proliferation, as both the total and reduced ASA content decreased in the later cultural stages.

GR is a flavinase which catalyzes the formation of GSH from GSSG. GR also plays a crucial role in maintaining cellular redox homeostasis by scavenging intracellular ROS. Previous studies have found that GR activity increases when plants are subjected to stressful environmental conditions, including exposure to ozone, drought, heavy metals, bright light, salt, and low temperatures. Additionally, the GR activity of stress-tolerant plant species remains high during stress exposure, but drops in susceptible species [37]. Plants adapted to high temperatures (*Eupatorium adenophora*) have been found to upregulate GR activity when exposed to low temperatures, but not high temperatures [38]. Contrarily, plants adapted to low temperatures (*Eupatorium odoratum*) show the opposite trend. In white spruce (*P. glauca*), the exogenous application of BSO during SE increased intracellular GR activity [39], while in oilseed rape (*Brassica napus*), the exogenous application of BSO during SE reduced intracellular GR activity [23]. In this work, we found that exogenous BSO decreased intracellular GR activity while exogenous GSH increased intracellular GR activity in F cell lines.

In the ASA-GSH cycle, ASA is converted by APX to monodehydroascorbic acid (MDHA) [40]. Changes in APX activity may be due either to changes in protein synthesis or to the kinetic properties of the enzyme [41,42]. APX activity is also responsive to environmental stress. For example, the activity of APX has been shown to increase under salt stress in she-oak (*Casuarina equisetifolia*) [43], pea (*Pisum sativum*) [44], citrus (*Citrus reticulata*) [45], and rice (*Oryza sativa*) [46]. Additionally, salt stress led to increased APX activity in new shoots of salt-tolerant potato (*Solanum tuberosum*) varieties, while APX activity decreased in salt-sensitive varieties. In oilseed rape (*B. napus*), APX activity was higher in embryos treated with exogenous BSO during the globular embryo period, and APX activity could be modified by exogenous GSH [23]. These results suggest that APX activity is controlled by intracellular GSH redox status, and that high levels of APX serve to reduce the total intracellular ASA content during the early development of BSO-treated embryos [23]. In this work, exogenous application of BSO was found to promote intracellular APX activity in F cell lines. Exogenous GSH inhibited intracellular APX activity in both S and F cell lines. It appears that in Korean pine, exogenous GSH promotes ASA synthesis and inhibits APX activity, leading to a decrease in DHA synthesis, while the opposite is true for BSO.

DHAR is a key enzyme in the ASA-GSH cycle, which promotes ASA regeneration and serves to protect cells against oxidative damage. Specifically, DHAR catalyzes the production of ASA from GSH, thus helping to regulate the ratio of intracellular ASA to DHA [47]. In this work, we found that the activity of DHAR was higher in S-CK than in the F-CK cell line. Exogenous application of BSO inhibited intracellular DHAR activity and GSH promoted intracellular DHAR activity in S cell lines. These results suggest that in Korean pine EC cells, exogenous GSH promotes intracellular synthesis of GSH, DHAR activity, and synthesis of ASA. When BSO is applied, the intracellular synthesis of GSH is inhibited, DHAR activity is decreased, and the synthesis of ASA is reduced.

## 4. Materials and Methods

### 4.1. EC Induction and Proliferation Assays

Two Korean pine cell lines, one with high proliferative potential 001#-001 (F) and one with low proliferative potential 001#-010 (S), were revived and cultured after ultra-low temperature preservation. EC induction was performed using mLV [48,49] medium with 4 g/L Gelrite, 500 mg/L L-glutamine, 30 g/L sucrose, 0.5 g/L hydrolyzed casein, 2 mg/L NAA, and 1.5 mg/L 6-BA. EC proliferation was performed using mLV medium with 4 g/L Gelrite, 500 mg/L L-glutamine, 30 g/L sucrose, 0.5 g/L hydrolyzed casein, 0.5 mg/L 6-BA, and 1 mg/L 2,4-D. For GSH treatments (F-GSH and S-GSH), 0.5 mmol/L GSH was dissolved in cool, sterile water, adjusted to pH 5.8, and added to the medium by filter sterilization (0.22 μm pore size). For BSO treatments (F-BSO and S-BSO), 0.5 mmol/L BSO was dissolved in sterile water, cooled to 55–60 °C, and added to the medium by filter sterilization (0.22 μm pore size). Treatments without GSH and BSO were used as controls (F-CK and S-CK). Samples of each treatment (three or four replicates each) were collected after 0, 7, 14, 21, 28, and 35 days of proliferation in order to assess fresh and dry weight, and intracellular GSH and ASA content was determined.

All culture media in this study had a pH of 5.8 and were autoclaved (121 °C, 20 min) after pH adjustment. After the medium was sterilized and cooled to approximately 55–60 °C, Gln and ABA were added by filter sterilization (filter membrane pore size 0.22 μm). The medium was inoculated with MG as the explants and the cultures were cultured in the dark at 23 ± 2 °C.

### 4.2. Determination of Intracellular Antioxidant Content and Enzymatic Activity

The intracellular GSH, GSSG, and ASA content, and GR and APX activity, were determined using Nanjing Jiancheng assay kits (GSH A 006-2-1, GSSG A 061-1, ASA A 009-1-1, GR A 062-1-1, APX A 123-1-1: Nanjing Jiancheng Institute of Biological Engineering, Nanjing, China, http://www.njjcbio.com/, accessed on 1 August 2022). Intracellular DHA content and DHAR activity were determined using Beijing Solarbio assay kits (DHA BC 1240, DHAR BC 0665: Beijing Solarbio Science & Technology Co., China, https://www.solarbio.com/, accessed on 1 August 2022). The absorbance values of GSH and GSSG were determined by an enzyme marker, and the absorbance values of DHAR, ASA, DHA, GR, and APX were determined by spectrophotometry, according to kit instructions. Samples were prepared for each assay according to the following paragraphs. Methods were as follows:

To determine GSH content: 0.2 g of EC was mixed with 1.8 mL of PBS buffer, homogenized in an ice bath, and centrifuged at 2500 rpm for 10 min. To determine GSSG content: 0.5 g of EC was mixed with 1 mL of freshly prepared reagent 4, homogenized in an ice bath, and centrifuged at 3500 rpm for 10 min. The absorbance value ‘A1’ was read at 405 nm for 30 s. The absorbance value ‘A2’ was read at room temperature (25 °C) for 10 min. The GSSG content was calculated according to the formula method. Three replicates per treatment were carried out. The concrete method please refer to the introductions.The concrete method please refer to the instructions.

To determine ASA content: 0.2 g of EC was mixed with 0.5 mL of PBS buffer, homogenized in an ice bath, and centrifuged at 5000 rpm for 10 min. To determine DHA content: 0.1 g of the sample was mixed with 0.5 mL of extract, homogenized in an ice bath, and centrifuged at 16,000 rpm for 20 min at 4 °C. The absorbance value ‘A1’ was recorded at 10 s, and the absorbance value ‘A2’ was recorded at 130 s. The change (Δ) was calculated as A2-A1, and 0.5 μmol/mL DHA was used as the standard tube. Four replicates per treatment were carried out. The concrete method please refer to the introductions.

To determine GR activity: 0.2 g of EC was mixed with 1.8 mL of PBS buffer, homogenized in an ice bath, and centrifuged at 2500 rpm for 10 min. Absorbance was measured at 340 nm with a 1 cm optical diameter cuvette. The absorbance value ‘A1’ was recorded at 30 s and the absorbance value ‘A2’ was recorded at 2 min at 37 °C. Double-distilled water was used as the standard tube. Four replicates per treatment were carried out. The concrete method please refer to the introductions.

To determine APX activity: 0.3 g of the sample was mixed with 0.6 mL of buffer, homogenized in an ice bath, and centrifuged at 10,000 rpm for 10 min at 4 °C. The absorbance value ‘A1’ was recorded at 10 s, and the absorbance value ‘A2’ was recorded at 130 s. The change (Δ) was calculated as A2-A1. Double-distilled water was used as the standard tube. Three replicates per treatment were carried out. The concrete method please refer to the introductions.

To determine DHAR activity: 0.6 g of the sample was added to 0.6 mL of extraction solution, homogenized in an ice bath, and centrifuged at 12,000 rpm for 10 min at 4 °C. The supernatant was added to the reagent according to the method of the kit, and the absorbance value at 412 nm of the enzyme-labeled instrument was measured. Three replicates per treatment were carried out introductions.

### 4.3. qRT-PCR Validation Assay

Transcriptomic sequencing revealed the presence of 10 differentially expressed genes between the F and S cell lines, and these genes were used for qRT-PCR validation. The Korean pine AAE7 gene (acyl-activating enzyme 7, c154707.graph_c1) was used as an internal reference gene. The AAE7 gene was selected because it is stable, with gene activity being largely independent of external interference. Transcriptomic sequencing revealed the presence of 10 differentially expressed genes between the F and S cell lines, and these were used for qRT-PCR validation. Primer sequences were synthesized by Beijing Bemec Biotechnology Co., (Beijing, China), and can be found in Table 1. RNA OD values were detected using an ultra-micro nucleic acid protein assay (ScanDrop 100, Analytik Jena, Germany). Reverse transcription was performed using a TRUEscript 1st Strand CDNA Synthesis Kit (Aidlab Biotechnologies Co., Beijing, China). The expression levels of target genes were determined using the 2-ΔΔCt calculation method, with three biological replicates per treatment.

### 4.4. Calculations and Statistical Analyses

Experimental calculations, including proliferation parameters, were performed using Excel 2003 (Microsoft, United States, Redmond, WA, USA). A one-way analysis of variance (ANOVA) and Duncan’s multiple comparisons tests were performed using SPSS 19 (IBM, New York, NY, USA). Data concerning percentages were arcsine-transformed before analysis. Graphs were constructed in Sigma Plot 12.0 (Systat, Richmond, CA, USA). Proliferation parameters were calculated according to the following equations:$$\mathrm{Relative}\;\mathrm{water}\;\mathrm{content}\;(\%)=\frac{\left(\mathrm{fresh}\;\mathrm{weight}\;-\;\mathrm{dry}\;\mathrm{weight}\right)}{\mathrm{fresh}\;\mathrm{weight}}\times 100$$

## 5. Conclusions

The study confirmed that exogenous GSH promotes the proliferation of Korean pine EC. This is because the exogenous application of GSH increased the intracellular contents of T-GSH, GSH, GSSG, T-ASA, and ASA and the ratios of GSH: T-GSH and ASA: T-ASA in both F and S cell lines. It appears that the application of exogenous GSH promotes a reducing cultural environment, which is beneficial to EC proliferation in Korean pine. By helping to reveal the mechanism whereby GSH regulates redox homeostasis in Korean pine EC cells, we have laid the foundation for a large-scale breeding of Korean pine somatic embryogenesis technology system.

## Figures and Tables

**Figure 1 plants-11-02586-f001:**
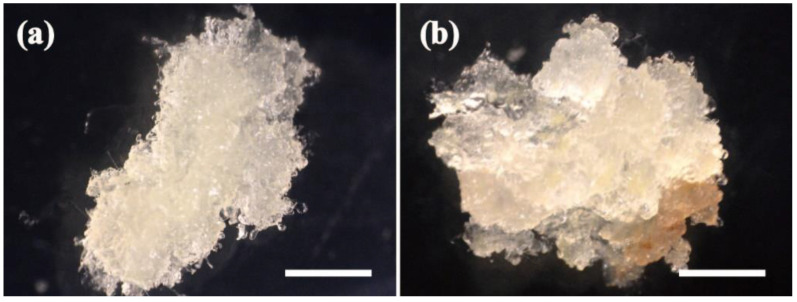
Macrostructure observation of embryogenic callus of Korean pine F and S cell lines. Note: (**a**): F cell line, (**b**): S cell line, bar = 1 cm.

**Figure 2 plants-11-02586-f002:**
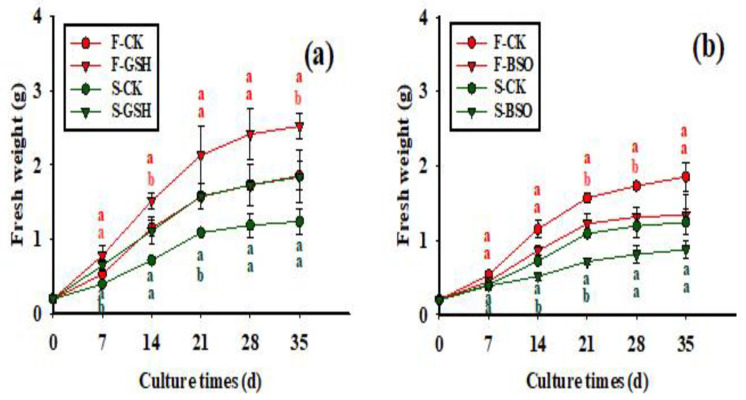
Effects of exogenous application of GSH and BSO on Korean pine EC FW. Note: (**a**) shows the CK and GSH treatment, (**b**) shows the CK and BSO treatment. Different lowercase letters at the same culture time indicate significant differences (*p* ˂ 0.05). Red lowercase letters indicate F cell line GSH, CK, BSO treatment, and green lowercase letters indicate S cell line GSH, CK, BSO treatment. The average EC weight of five Petri dishes was counted for each treatment. ANOVA and Duncan’s test were performed on the data (mean ± se) in the figure. F cell line control treatment (F-CK); F cell line exogenously supplemented with GSH (F-GSH); F cell line exogenously supplemented with BSO (F-BSO); S cell line control treatment (S-CK); S cell line exogenously supplemented with GSH (S-GSH); S cell line exogenously supplemented with BSO (S-BSO), Figure 3, Figure 4, Figure 5, Figure 6 and Figure 7 are the same as Figure 2.

**Table 1 plants-11-02586-t001:** qRT-PCR primer sequences.

Unigenes	Forward Primer	Reverse Primer
c154707.graph_c1 (Internal reference)	CAGCTCAGCTTGAAATGA	ACGAGTACTGTGAGGAAA
c180344.graph_c0	CGACATGGTTCCAAACTA	GACCCTGAGAAAATTGTTG
c187964.graph_c1	TGGAAACCCTGTTTGTGAA	GCAATGGCTCGGTCATAT
c188830.graph_c0	CTCCTAGCTCTCAGGTCAT	CTTTGCGATGAGTTGTGC
c181305.graph_c0	CTGGTTGGTTACACGTTCC	TGTGCATGGATAGACCGAT
c187768.graph_c0	ATGGACATGAAGCATTGC	CGGTGATTGGTTACAACC
c183265.graph_c0	TTCTCAATCCGAATCTGGAA	AGGAGATGTCACCGTAGTA
c190620.graph_c2	GTGAAGCCGTGTCGAAGA	GGCGTTTCGTTTCACGTC
c187045.graph_c0	GCAAGAGCTACAGATCAA	GGACGAAATCGTATTCAAC
c133544.graph_c0	AGGGGATGAATTCAGCATA	CGCATTCACATACTTTCGA
c187027.graph_c1	TGATGAAGGCTTCCACAGA	CAAGTGCTATGCGAACCC

## Data Availability

Not applicable.

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
