# Peer review of "Exogenous Glutathione Promotes the Proliferation of Pinus koraiensis Embryonic Cells and the Synthesis of Glutathione and Ascorbic Acid"

_plants, 2022, doi:10.3390/plants11192586_

Round 1
Reviewer 1 Report
Dear Editor dear Author,
Here is my evaluation of the article Plants-1880932:
The manuscript present the influence of glutathione on Pinus koraiensis SE and especially CE proliferation.
The manuscript presents the influence of exogenous application of glutathione on growth (proliferation rate) of two lines of somatic embryos (SE), and embryonic callus (EC) tissue of korean pine Pinus koraiensis). They found that the exogenous application of GSH increased the intracellular levels of GSH, total GSH (T-GSH), oxidized GSH (GSSG), ascorbic acid (ASA), total ASA (T-ASA), and the ratios of GSH:T-GSH and ASA:T-ASA in both F and S cell lines.
The research topic is suitable for publication, but the manuscript has some shortcomings that need to be improved and require extensive revision before publication!
General comment:
- The Latin name of Korean pine (Pinus koraiensis) needs to be italicized: Korean pine (Pinus koraiensis),
- Please place all figures and tables after the paragraph in which they were first mentioned
- The figures are not self-explanatory. Numbers of explants are missing. Detailed description of statistics is missing. Abbreviations have not been described.
- Check the references, give them as numbers in the manuscript!
- I find it hard to believe that both authors working on Korean pine (Peng, 2020; Peng, 2021) are only mentioned in the discussion.
- I find it hard to believe that there are no references to work on other pine genera or other gymnosperms.
Comments:
Abstract:
Line 26:
The role of L-buthionine-sulfoximine (BSO) is not explained: explain it more detail or omit it from abstract.
Line 33
Conclusion “By helping to reveal the mechanism whereby GSH regulates redox homeostasis in Korean pine EC 34 cells, we have laid the foundation for improving the efficiency of SE in Korean pine, and perhaps 35 other woody species.” Is somehow too speculative.
Introduction:
Line 51
Add a new paragraph after “Korean pine (Pinus koraiensis Sieb. et Zucc.) is a conifer of particular economic importance, producing high-quality wood for use in construction, furniture…….”
Line 73: The purpose and objective are not clearly stated.
Results:
The position of all figures must be after the paragraph in which they were first mentioned.
The illustrations are not self-explanatory. The number of explants is missing. A detailed description of the statistics is missing. SE or SD?, average figures, post-hoc ?
Why are not both F- and S-lines presented in the same figure with statistical analysis?
Figures 2 and 3 were without statistical analysis.
Figure 4 and 5: We hardly distinguish between CK, GSH and BSO on line charts.
Relative water content (RWC) or water content (WC) of more than 1000? Please check the formula for calculation
Proliferation quality needs to be changed to FW or DW increase. Please check the difference between absolute growth rate (AR), relative growth rate (RGR) and relative increment (RI)!
Data on percent cell division and cell death are missing!
Please show some pictures of plant material.
Discussion:
I will read if data on promoting growth and development is presented or pure growth as FW and DW or with another parameter as absolute growth rate.
Out of 40 references only 7 are newer than 2010 and only 4 are from 2022-2021, none from 2022! I find it hard to believe that there are no more recent references dealing with the use of antioxidants in embryogenesis of the genus Pinus and others:
Hazubska-Przybył, T., Wawrzyniak, M. K., Kijowska-Oberc, J., Staszak, A. M., & Ratajczak, E. (2022). Somatic Embryogenesis of Norway Spruce and Scots Pine: Possibility of Application in Modern Forestry. Forests, 13(2), 155.
Material and methods:
In EC induction and proliferation assays I missing some references about culturing SE and CE.
What is mLV medium? Reference?
Please provide more data (website) about the supplier and assay kits used in this study: Nanjing Jiancheng Assay Kits (GSH A 006-2-1, GSSG A 061-1, ASA A 009-391 1-1, GR A 062-1-1, APX A 123-1-1: Nanjing Jiancheng Institute of Biological Engineering, 392 Nanjing, China).
Conclusion:
Is too speculative. Please focus on your results.
Author Response
Response to Reviewer 1 Comments Dear Reviewer, Our sincere thanks to you for the time and effort that you have put into reviewing our manuscript! We found all the comments very constructive and helpful, and have revised our manuscript according to all comments. Please find, below, our point-by-point response to the comments raised. Thank you for considering our revised manuscript! Point 1:The Latin name of Korean pine (Pinus koraiensis) needs to be italicized: Korean pine (Pinus koraiensis) Response 1: All Pinus koraiensis in the text have been changed to italics. Point 2:Please place all figures and tables after the paragraph in which they were first mentioned Response 2: All charts have been placed after the paragraph in which they are first mentioned. Point 3: The figures are not self-explanatory. Numbers of explants are missing. Detailed description of statistics is missing. Abbreviations have not been described. Response 3:The information of the figure taken, the data statistics, has been taken according to the reviewers' comments. Abbreviation information added (The average EC weight of five Petri dishes was counted for each treatment, ANOVA and Duncan's test were performed on the data in the figure. F cell line control treatment (F-CK); F cell line exogenously supplemented with GSH (F-GSH); F cell line exogenously supplemented with BSO (F-BSO); S cell line control treatment (S-CK); S cell line exogenously supplemented with GSH (S-GSH); S cell line exogenously supplemented with BSO (S-BSO), The following pictures is the same as this .) Point 4:Check the references, give them as numbers in the manuscript! Response 4: The number of references and word count have been given after the abstract. Point 5:I find it hard to believe that both authors working on Korean pine (Peng, 2020; Peng, 2021) are only mentioned in the discussion. Response 5: The study of Peng has been cited in the introduction, along with the latest study of Peng et al. 2022. Peng C, Gao F, Wang H, Tretyakova IN, Nosov AM, Shen H, Yang L. Morphological and Physiological Indicators for Screening Cell Lines with High Potential for Somatic Embryo Maturation at an Early Stage of Somatic Embryogenesis in Pinus Koraiensis. Plants. 2022; 11(14):1867. Point 6:I find it hard to believe that there are no references to work on other pine genera or other gymnosperms. Response 6: Research work on other coniferous species has been cited in the manuscript. Sun T, Wang Y, Zhu L, Liu X, Wang Q, Ye J. Evaluation of somatic embryo production during embryogenic tissue proliferation stage using morphology,maternal genotype,proliferation rate and tissue age of pinus thunbergii parl [J]. Journal of Forestry Research, 2022, 33(2), 10. Song Y , Gao X , Wu Y . Key Metabolite Differences Between Korean Pine (Pinus koraiensis) Seeds With Primary Physiological Dormancy and No-Dormancy[J]. Frontiers in Plant Science, 2021, 12:2512-. Hazubska-Przybył T, Wawrzyniak M K, Kijowska-Oberc J, Staszak A M, Ratajczak E. Somatic Embryogenesis of Norway Spruce and Scots Pine: Possibility of Application in Modern Forestry. Forests,2022, 13(2), 155. Rosvall O. Using Norway spruce clones in Swedish forestry: general overview and concepts[J]. Scandinavian Journal of Forest Research, 2019, 34(943):1-20. Peng C, Gao F, Wang H, Tretyakova IN, Nosov AM, Shen H, Yang L. Morphological and Physiological Indicators for Screening Cell Lines with High Potential for Somatic Embryo Maturation at an Early Stage of Somatic Embryogenesis in Pinus Koraiensis. Plants. 2022; 11(14):1867.

Reviewer 2 Report
“Exogenous glutathione promotes the proliferation of Pinus koraiensis embryonic cells and the synthesis of glutathione and ascorbic acid” has been thoughtfully reviewed and some comments are suggested:
General comments
1. The manuscript has scientific relevance, presents an appropriate introduction, well-described material and methods, consistent results, and an adequate discussion. However, some points need to be considered and revised.
2. One of the central premises of the work was to stimulate the proliferation of embryogenic cultures, which was achieved. However, this aspect is not contemplated in the abstract and in the conclusions and must be included.
3. Acronyms must be spelled out in full the first time they are used.
4. Authors should avoid using "significantly different";. If the data were subjected to statistical analysis and that analysis indicated differences, it is not necessary to use the term "significant".
5. Figures 1, 2, 3, and 6 need to be reformulated to a graph pattern similar to that shown in Figures 4 and 5. Lines joining data should be used when presenting time series.
6. All legends for figures and tables must be completely self-explanatory. Please review the legends, especially figure 7.
7. In the topic "4.3. qRT-PCR validation assay" the authors need to include the source on which they based the sentence "Transcriptomic sequencing revealed the presence of 10 differentially expressed genes between the F and S cell lines, and these were used for qRT-PCR validation";
8. Authors should better explain the calculations used for proliferation parameters. Does the fresh weight of embryogenic cultures before proliferation correspond to time 0 or the immediately preceding measurement?
Author Response
Response to Reviewer 3 Comments
Dear Reviewer,
Our sincere thanks to you for the time and effort that you have put into reviewing our manuscript! We found all the comments very constructive and helpful, and have revised our manuscript according to all comments. Please find, below, our point-by-point response to the comments raised.
Thank you for considering our revised manuscript!
Point 1: One of the central premises of the work was to stimulate the proliferation of embryogenic cultures, which was achieved. However, this aspect is not contemplated in the abstract and in the conclusions and must be included.
Response 1: GSH-promoted EC proliferation has been included in the abstract.
Point 2: Acronyms must be spelled out in full the first time they are used.
Response 2: Already adding the full spelling of acronyms, such as: fresh weight (FW); dry weight (DW); reactive oxygen species (ROS)
Point 3:Authors should avoid using "significantly different";. If the data were subjected to statistical analysis and that analysis indicated differences, it is not necessary to use the term "significant".
Response 3: The description of the text involving “significantly” has been revised.
Point 4:Figures 1, 2, 3, and 6 need to be reformulated to a graph pattern similar to that shown in Figures 4 and 5. Lines joining data should be used when presenting time series.
Response 4: We have changed figures 1, 2, 3, and 6 to the line graph.
Point 5:All legends for figures and tables must be completely self-explanatory. Please review the legends, especially figure 7.
Response 5: The figure captions for figures 1 through 7 have been revised.
Point 6:In the topic "4.3. qRT-PCR validation assay" the authors need to include the source on which they based the sentence "Transcriptomic sequencing revealed the presence of 10 differentially expressed genes between the F and S cell lines, and these were used for qRT-PCR validation";
Response 6: "Transcriptomic sequencing revealed the presence of 10 differentially expressed genes between the F and S cell lines, and these genes were used for qRT-PCR validation "has been added in 4.3.
Point 7:Authors should better explain the calculations used for proliferation parameters. Does the fresh weight of embryogenic cultures before proliferation correspond to time 0 or the immediately preceding measurement?
Response 7: The proliferation weight in figures 1 and 2 have been changed to fresh and dry weight.
Remaining modified sections: Figures 1-7 in the text were changed to color graphs, and the symbols of each data point were enlarged, and the GSH, CK, and BSO folds were changed to different colors, with the exogenous GSH treatment changed to red and the exogenous BSO treatment changed to green, making it easier to distinguish the CK, GSH, and BSO treatments.

Round 2
Reviewer 1 Report
Dear Editor dear Author,
Here is my additional evaluation of the article Plants-1880932:
The research topic is suitable for publication, but the manuscript has some shortcomings that were not improved and require further revision before publication:
The figures are not self-explanatory. Numbers of explants are missing. Detailed description of statistics is missing in Figure 3, 4, 5, 6. The number of explants is missing. A detailed description of the statistics is missing. SE, average figures, post-hoc ?
Why are not both F- and S-lines presented in the same figure with statistical analysis?
Relative water content (RWC) or water content (WC) of more than 1000? Please check the formula for calculation.
Several results are shown as percentages (Fig 4). How they tused these percentages in calculation of significant differences?
How they calculated statistics? What kind of statistical test was used? Which post hoc was used?
Figure 5, 6: Statistics of a, b, g, h is missing.
With regard, reviewer
Author Response
Dear Reviewer,
Our sincere thanks to you for the time and effort that you have put into reviewing our manuscript! We found all the comments very constructive and helpful, and have revised our manuscript according to all comments. Please find, below, our point-by-point response to the comments raised.
Thank you for considering our revised manuscript!
Point 1:The figures are not self-explanatory. Numbers of explants are missing. Detailed description of statistics is missing in Figure 3, 4, 5, 6. The number of explants is missing. A detailed description of the statistics is missing. SE, average figures, post-hoc ?
Response 1: The Figure 3, 4, 5, 6 has been supplemented with information on fetching, data statistics, and abbreviation information.
For example:Note: Different lowercase letters at the same culture time indicate significant differences (p ˂ 0.05). The average EC weight of five Petri dishes was counted for each treatment, ANOVA and Duncan's test were performed on the data (Mean ± se) in the figure.
Point 2: Why are not both F- and S-lines presented in the same figure with statistical analysis?
Response 2: All Figure F and S cell lines in the manuscript have been statistically analyzed in the same figure, as detailed in the manuscript.
Point 3: Relative water content (RWC) or water content (WC) of more than 1000? Please check the formula for calculation.
Response 3: The absolute water content in the manuscript has been changed to relative water content.
Point 4: Several results are shown as percentages (Fig 4). How they tused these percentages in calculation of significant differences? How they calculated statistics? What kind of statistical test was used? Which post hoc was used?
Response 4: Data concerning percentages were arcsine-transformed before analysis (4.4 Calculations and Statistical analyses). A specific method of statistical analysis has been added to Figure 3~7, For example: Different lowercase letters at the same culture time indicate significant differences (p ˂ 0.05). The average EC relative water content of five Petri dishes was counted for each treatment, ANOVA and Duncan's test were performed on the data (Mean ± se) in the figure.
Point 5: Figure 5, 6: Statistics of a, b, g, h is missing.
Response 5: Figure 5, 6: a, b, g, h in the manuscript have been added to the graph statistical analysis.
Reviewer 2 Report
The authors have addressed all my queries and, in my opinion, the ms can be accepted as it is.
Author Response
Dear Reviewer,
Our sincere thanks to you for the time and effort that you have put into reviewing our manuscript!
Thank you for considering our revised manuscript!